# Optical control of exciton spin dynamics in layered metal halide perovskites via polaronic state formation

Sean A. Bourelle[1,8], Franco V. A. Camargo[2,8], Soumen Ghosh [3,8], Timo Neumann[1,4], Tim W. J. van de Goor [1], Ravichandran Shivanna [1,5], Thomas Winkler [1,6], Giulio Cerullo[2,3 ✉] & Felix Deschler[4,7 ✉]

One of the open challenges of spintronics is to control the spin relaxation mechanisms. Layered metal-halide perovskites are an emerging class of semiconductors which possess a soft crystal lattice that strongly couples electronic and vibrational states and show promise for spintronic applications. Here, we investigate the impact of such strong coupling on the spin relaxation of excitons in the layered perovskite $BA_2FAPbI_7$ using a combination of cryogenic Faraday rotation and transient absorption spectroscopy. We report an unexpected increase of the spin lifetime by two orders of magnitude at 77 K under photoexcitation with photon energy in excess of the exciton absorption peak, and thus demonstrate optical control over the dominant spin relaxation mechanism. We attribute this control to strong coupling between excitons and optically excited phonons, which form polaronic states with reduced electron-hole wave function overlap that protect the exciton spin memory. Our insights highlight the special role of exciton-lattice interactions on the spin physics in the layered perovskites and provide a novel opportunity for optical spin control.

[1] Cavendish Laboratory, University of Cambridge, J J Thomson Avenue, Cambridge CB3 0HE, UK. [2] Istituto di Fotonica e Nanotecnologie-CNR, Piazza Leonardo da Vinci 32, 20133 Milano, Italy. [3] Dipartimento di Fisica, Politecnico di Milano, Piazza Leonardo da Vinci 32, 20133 Milano, Italy. [4] Walter-Schottky-Institute, Physics Department, Technical University Munich, Am Coulombwall 4, Garching, Germany. [5] Department of Physics, Indian Institute of Technology Madras, Chennai 600036, India. [6] Department of Physics and Astronomy, Aarhus University, 8000 Aarhus C, Denmark. [7] Present address: Physikalisch-Chemisches Institut, Universität Heidelberg, Im Neuenheimer Feld 229, 69120 Heidelberg, Germany. [8] These authors contributed equally: Sean A. Bourelle, Franco V. A. Camargo, Soumen Ghosh. ✉email: giulio.cerullo@polimi.it; deschler@uni-heidelberg.de

Materials for spintronics devices, which aim to utilise the spin degree of freedom for data storage and processing, require both efficient spin injection and long spin lifetimes[1–5]. However, traditionally, these two requirements are associated with quite different material properties. On the one hand, strong spin–orbit coupling (SOC) enables efficient spin-charge conversion[6], while on the other hand, it typically increases the rate of spin relaxation[7,8]. Therefore, suppressing spin depolarisation in the presence of strong SOC is an ongoing challenge for semiconductor spintronics[9] that could be addressed by metal halide perovskites[10–13].

In contrast to conventional semiconductors, metal halide perovskites exhibit strong coupling between the electronic and vibrational degrees of freedom[14,15]. Studies have shown that dynamic strain fields arise in response to photoexcitation[16], and that the low mean free path of phonons implies highly localised lattice vibrations[17]. Within the layered metal-halide perovskites, the term 'exciton–polaron' has been introduced to describe the coupling between excitons and lattice vibrations[18–21], while further studies have indicated the formation of polaron states[22] which are not well described by the Fröhlich Hamiltonian[23] and may require a significantly more complex theoretical treatment[24]. For simplicity, in this paper, we refer to the former simply as excitons and the latter as polaronic states.

Layered metal-halide perovskites are quasi-2D self-assembled quantum wells of the form $A_2B_{n-1}PbI_{3n+1}$, where $n$ is the number of lead-iodide unit cells between the organic spacer cations (A). These lead-iodide cages define the electronic orbitals and give rise to strong SOC[25], which is typically utilised for spin-to-charge conversion within spintronic devices[6]. Reducing $n$, i.e. reducing the quantum-well width, has been shown to increase exciton–phonon coupling[26,27]. On the other hand, the small polar cations (B) are expected to increase the exciton–phonon coupling strength[28], to couple their dipole moment to that of the excitons[29], and to induce a localised dynamic disorder at elevated thermal energies when its rotational barrier is overcome[17]. These considerations suggest that exciton–phonon coupling may be strongest for bilayer ($n = 2$) perovskites—the narrowest wells containing intralayer cations.

While exciton–lattice coupling has received significant interest in these materials[14,15,18–21,30,31], it remains an open question how it will impact the spin dynamics. Interestingly, it has been calculated that polaron formation in the presence of strong SOC and broken inversion symmetry[32] significantly affects spin relaxation and transport efficiency[33]. Yet, while previous studies of spin relaxation within 2D perovskites[11,34–37] have reported dependence on layer number[32], excitation density[38] and chemical composition[39], the impact of strong exciton–phonon coupling on spin depolarisation is still unexplored.

Here, we employ a set of ultrafast optical spectroscopies from ambient to cryogenic temperatures and report that optical excitation with excess photon energy increases the cryogenic spin lifetime by two orders of magnitude due to a change in the dominant spin-relaxation mechanism. Using time-resolved Faraday rotation (FR) and transient absorption (TA) spectroscopy we propose that the ultrafast formation of polaronic states is the underlying mechanism that protects spin by reducing the rate of spin precession. Our results demonstrate that the strong electron-phonon coupling in 2D perovskites sets them apart as a unique material system in which the dominant spin-relaxation mechanism can be optically selected and a novel regime of polaronic spin dynamics can be studied. This unexpected ability to encode the photoexcitation energy into the spin lifetime can be a promising tool for future spintronic–photonic devices.

## Results

**Excitation-energy dependence of the spin lifetime.** Polycrystalline films of the 2D-layered hybrid perovskite $BA_2FAPbI_7$, (BA = butylammonium, FA = formamidinium, Pb=lead, I = iodide), whose structure is sketched in Fig. 1a, are prepared by spin coating (see 'Methods') and optical measurements are then performed on multiple films to ensure consistency. Steady-state absorption and photoluminescence (PL) spectra are taken at room temperature and show a clear excitonic transition with no discernible contribution from 2D perovskites of $n \neq 2$ (Fig. 1b and Supplementary Fig. 1).

We perform FR measurements, which track the decay of exciton spin polarisation, using pulses of 40 meV full-width-half-maximum bandwidth and 0.1 ps duration (Fig. 1b, c). Optical excitation using circularly polarised light generates excitonic states with polarised total angular momentum $|\pm1\rangle$[38]. These states then depolarise via lattice and exciton interactions, with only the former being relevant at the low exciton densities corresponding to natural sunlight illumination levels. Therefore, we perform all measurements at low fluences (exciton density around $10^{16}$ cm$^{-3}$) for which many body interactions do not play a significant role (see fluence-dependent data reported in Supplementary Fig. 2).

Motional narrowing type spin depolarisation, due to the exchange driven Maialle-Silva-Sham (MSS)[40] or to the D'yakonov-Perel (DP)[41] mechanism, has been reported within $BA_2FAPbI_7$ for moderate carrier densities following photoexcitation that is resonant with the lowest energy exciton[34]. In this regime, spin dephasing occurs due to precession around an effective magnetic field. As phonon scattering leads to random changes in the direction of the spin–orbit interaction, increased phonon scattering averages out spin dephasing and yields a longer spin lifetime[42]. To our knowledge, the spin dynamics that follow photoexcitation with excess energies beyond 50 meV have never been investigated, presumably because it is known that under low fluences excess energy is quickly dissipated through carrier-phonon scattering[43–45]. However, precisely because spin dephasing mechanisms are linked to phonons, the question of whether photoexcitation with excess energy leads to new lattice distortions that influence spin lifetimes is an intriguing one.

Figure 1d shows time-resolved FR of $BA_2FAPbI_7$ at room temperature (dashed lines) and at 77 K (solid lines) after photoexcitation either in resonance with the exciton (2.17 eV, red lines) or with excess energy (2.43 eV, blue lines). At room temperature, we observe bi-exponential depolarisation in both cases which is dominated by the long $\tau_2$ component of ~3 ps, with little dependence on pump photon energy. At 77 K, on the other hand, a striking difference is observed: excitation with 2.17 eV photons leads to a rapid spin decay with a monoexponential lifetime of ~0.3 ps, while excitation with 2.43 eV photons leads to a spin lifetime that is two orders of magnitude longer (~40 ps). This greater than 100-fold increase cannot be explained by increased phonon scattering within the motional narrowing regime, since the spin lifetime is significantly shorter at room temperature under the same 2.43 eV excitation. Therefore, we need to perform a more detailed investigation of the impact of phonons on spin relaxation.

**Analysis of dominant spin-relaxation mechanisms.** To isolate the effects of phonon scattering on the spin-relaxation dynamics, we perform temperature-dependent measurements. Figure 2a shows the FR at different temperatures following 2.17 eV excitation and reveals that the spin polarisation lifetime increases with temperature. Figure 2b shows the results of bi-exponential

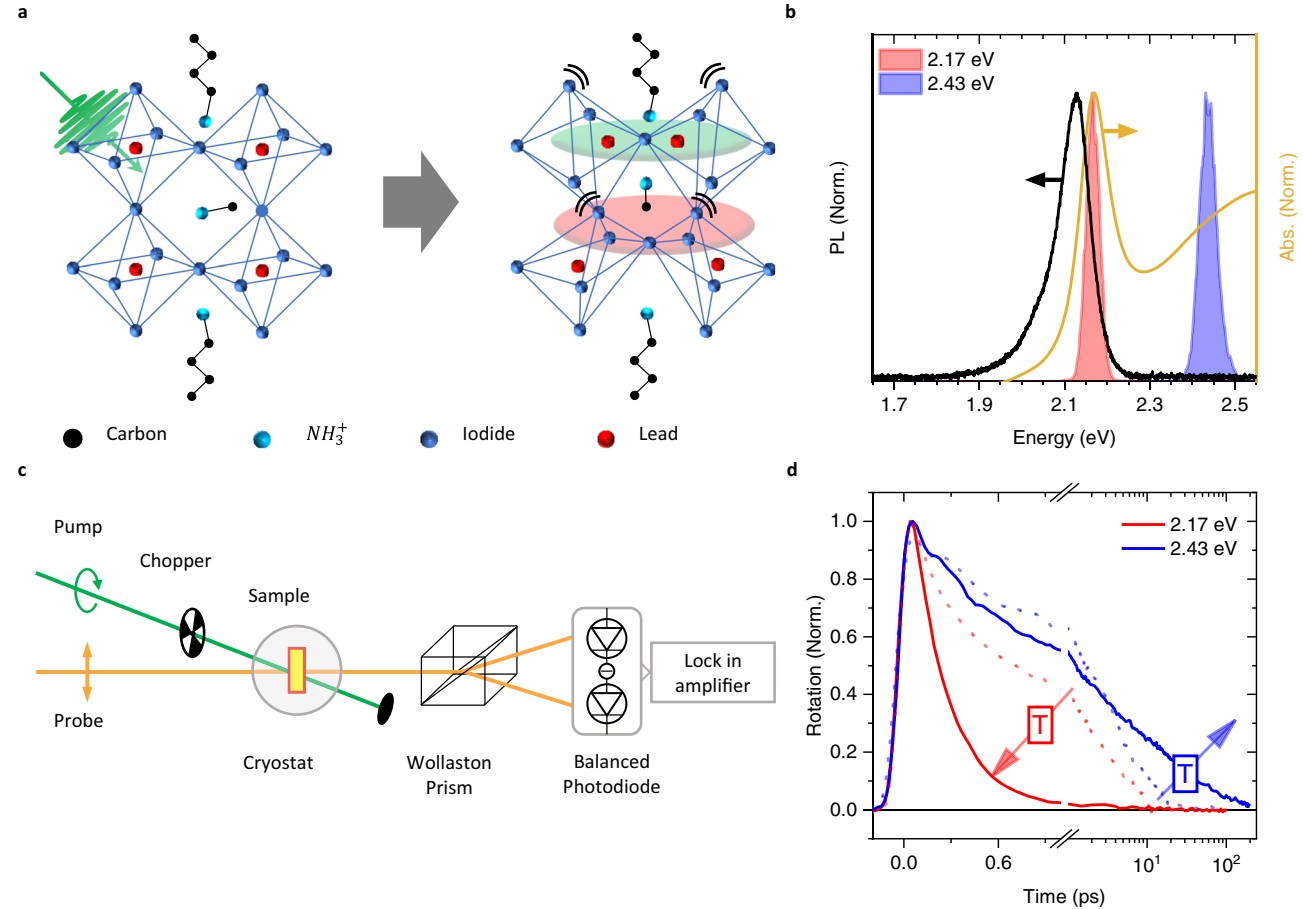

**Fig. 1 Material characterisation and Faraday rotation measurements. a** Sketch of the 2D perovskite BA$_2$FAPbI$_7$ structure and of lattice deformation associated with polaronic states. **b** Absorption (yellow line), and photoluminescence spectra (black line) of BA$_2$FAPbI$_7$ at 77 K. Spectra of the pump pulses used in TA and FR measurements are shown as red and blue shaded lines, respectively. **c** Experimental setup for FR measurements using a balanced photodetector (BPD). Probe photon energy is set to 1.91 eV, below the material absorption onset. **d** Temperature-dependent FR dynamics (spin polarisation as a function of time) following excitation with 2.17 eV and 2.43 eV pump photon energies at 77 K (solid lines) and at room temperature (dashed lines). All measurements had an exciton density smaller than 5 × 10$^{16}$ cm$^{-3}$ (see Supplementary Fig. S2). Lowering the temperature under 2.17 eV excitation decreases the spin lifetime, while under 2.43 eV excitation it leads to an increase.

fits to this data and confirms that the spin relaxation is dominated by a motional narrowing contribution that leads to a short spin polarisation lifetime, $\tau_1$, when the phonons are frozen out.

Within a motional narrowing regime, an increase in the temperature can lead to both a higher momentum scattering rate, $\Gamma$ *and* a faster spin precession, $\Omega(\boldsymbol{K})$, around an effective magnetic field if there is a non-zero average centre of mass wavevector (Supplementary Note 1). The temperature dependence of these two contributions determines the temperature dependence of the total spin-polarisation lifetime:

$$\tau_S \propto \frac{\Gamma}{\Omega(\boldsymbol{K})^2} \qquad (1)$$

To determine the temperature dependence of $\Gamma$, we measure PL spectrum under 2.17 eV excitation as a function of temperature (Supplementary Fig. 3) and use the PL linewidth, $\Gamma_{PL}$, to obtain $\Gamma \propto \Gamma_{PL}$ (Supplementary Eq. (S11)). We find that both $\tau_1$ and $\Gamma_{PL}$ follow the same power-law dependence of $\tau_1 \propto \Gamma_{PL} \propto T^{5/2}$ below 240 K (blue dots and black squares in Fig. 2b). This temperature dependence is consistent with that reported for 3D bulk perovskites[46]. Beyond 240 K (grey regions of Fig. 2b, d), the material undergoes a phase transition which may be linked to an

order−disorder phase transition[47] as seen in BA$_2$PbI (Supplementary Fig. 4) that leads to a deviation from this power law. Such a linear proportionality between spin lifetime and the momentum scattering rate (Supplementary Fig. 5) suggests from Eq. (1) that the precession term, $\Omega(\boldsymbol{K})$, is nearly temperature-independent. This result is consistent with previous studies which have shown cooling through a vibrational manifold of distinct exciton states[23,30,48,49], and have suggested a narrow spread of K-values such that that high-$\boldsymbol{K}$ excitons are unlikely to exist[45]. Interconversion within this manifold allows the average exciton energy to increase without a corresponding increase in exciton momentum. Alternatively, the temperature independence of $\Omega(\boldsymbol{K})$ can be explained by a zero average centre of mass wavevector.

We now consider the second contribution to the bi-exponential fit in Fig. 2a, which reveals a longer spin lifetime, $\tau_2$, that— contrary to $\tau_1$—decreases with increasing temperature. The amplitude of $\tau_2$ is zero below 180 K, Fig. 2b, but shows a greater contribution towards higher temperatures, indicating the onset of a new spin-relaxation mechanism as thermal phonons increase in number.

To investigate this new mechanism, we introduce optically excited phonons by repeating the FR temperature series with a photoexcitation energy of 2.43 eV. The spin polarisation now

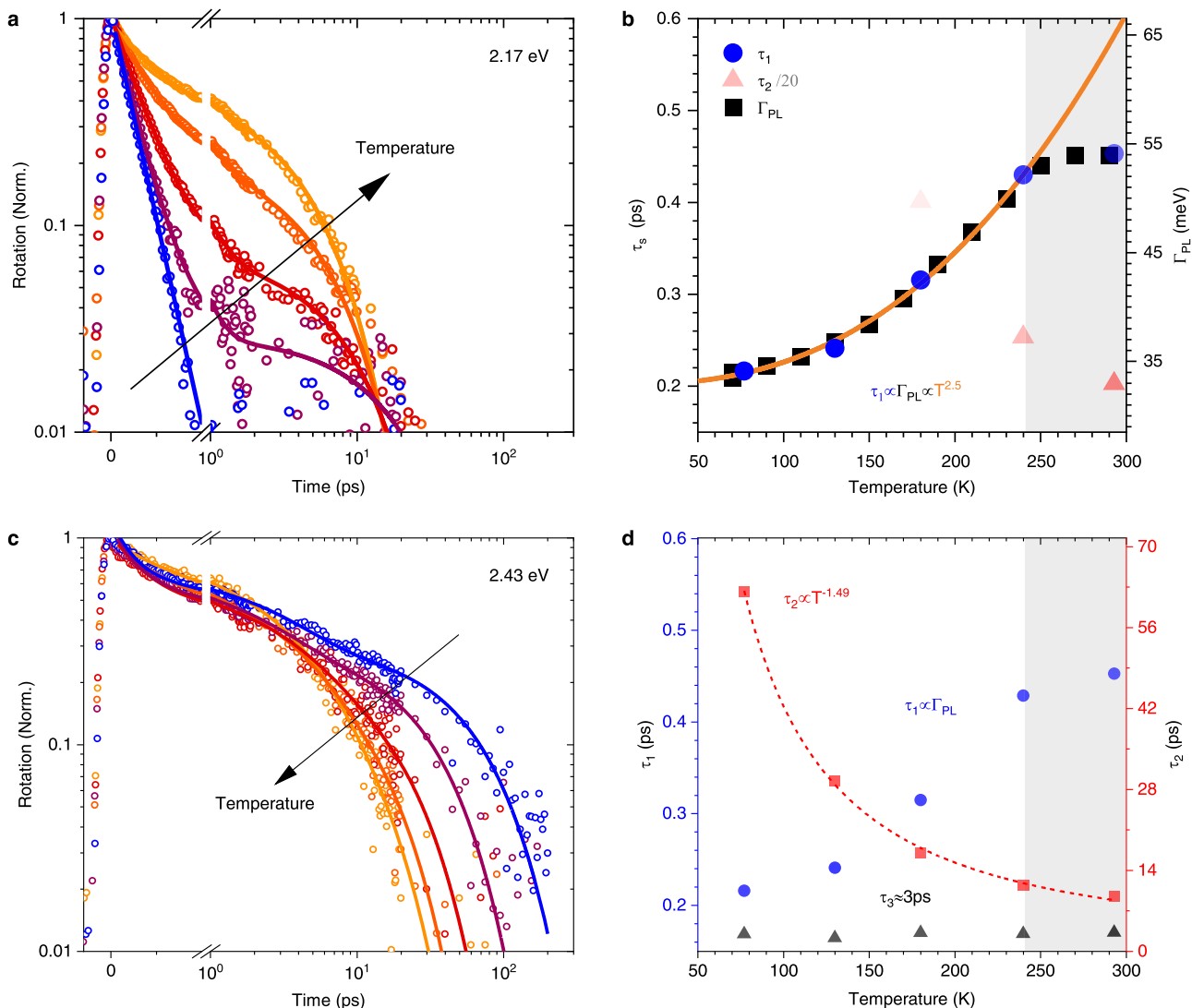

**Fig. 2 Faraday rotation signal as a function of temperature: from blue (77 K) through (130 K, 180 K, 240 K) to orange (293 K). a** FR under 2.17 eV excitation and exciton density $<3 \times 10^{16}$ cm$^{-3}$ (points) and fits to the bi-exponential decays (solid lines). **b** Time constants, $\tau_1$ and $\tau_2$, obtained from a fit to (**a**) with symbol transparency scaled to their relative amplitudes, and the full-width-half-maximum of the PL lineshape, $\Gamma_{PL}$, plotted as a function of temperature (Supplementary Fig. 3). $\tau_2$ is scaled by 1/20 to plot it on the same scale. **c** FR with excitation energy of 2.43 eV and exciton density $<0.67 \times 10^{16}$ cm$^{-3}$ (points) and fits to the tri-exponential decays (solid lines). **d** Time constants obtained from a fit to (**c**) with transparency scaled to amplitude (~33% amplitude of each component for all temperatures). Grey boxes in panels **b** and **d** indicate a new crystalline phase following a phase transition at 240 K (Supplementary Fig. S4).

persists beyond 200 ps, with a lifetime that, again, decreases with increasing temperature, Fig. 2c. This agrees with the previous observation that there exist states, formed in the presence of phonons, for which phonon scattering instead induces depolarisation. We find that the kinetics require a tri-exponential fit to describe the decay, Fig. 2d. As hot-carrier relaxation takes place on ~0.4 ps timescales at low carrier densities[43–45], the fast spin relaxation observed in Fig. 2a, b is expected to persist to some degree. Therefore, we constrain the first component of the triexponential fit to the same $\tau_1$ values found under 2.17 eV excitation and find that it accounts for about one-third of the amplitude at all temperatures. The third component $\tau_3$, yields a temperature-independent value of 2.9 ± 0.4 ps, which is consistent with the room-temperature spin lifetime, and could be attributed to some local heating of the lattice. Importantly, the second component, $\tau_2$, yields an inverse temperature dependence, with ~60 ps lifetime at 77 K, which cannot be explained by motional narrowing. A similar temperature dependence

(although independent of pump energy) is observed for 3D perovskite structures[50], with a spin-relaxation time of ~40 ps and $\tau_s \propto T^{-3/2}$. We find excellent agreement with this dependence and note that, like in bulk perovskites[50], this deviates from the $\tau_s \propto T^{-7/2}$ expected by the Elliott–Yafet (EY) spin-relaxation mechanism (Supplementary Note 1).

Thus, we conclude that photoexcitation of BA$_2$FAPbI$_7$ with excess energy (Supplementary Fig. S6) generates new electronic states which are less susceptible to fast motional narrowing spin relaxation. Consequently, the dominant spin-relaxation mechanism changes to one which resembles that of the 3D perovskites[50], and remains to be determined.

**Coherent exciton–phonon coupling and polaron formation.** To examine the strong exciton–phonon coupling under excess excitation energy and the possible formation of polaronic states in BA$_2$FAPbI$_7$, we perform TA measurements. Figure 3a, b shows the TA maps at 77 K following photoexcitation at 2.17 eV and

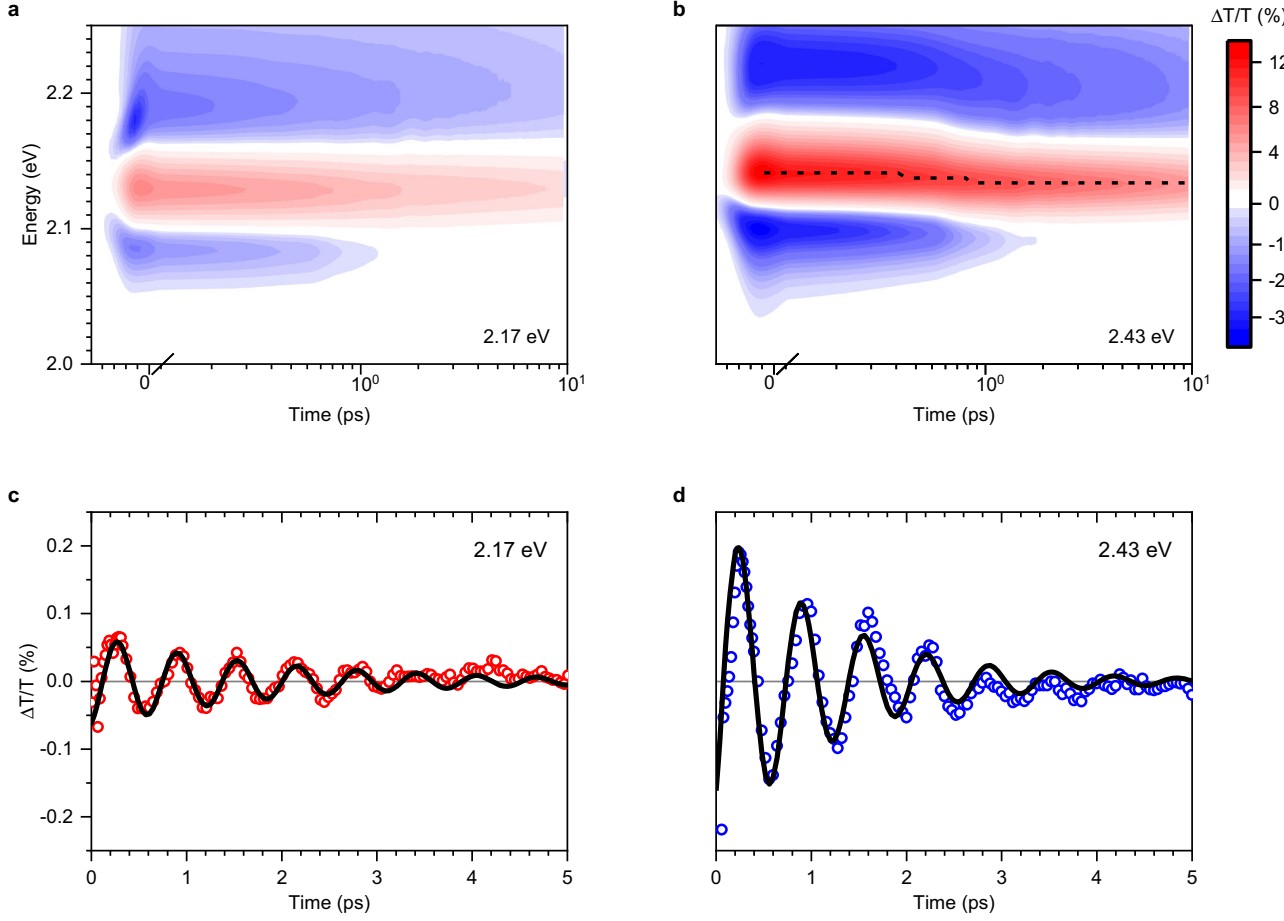

**Fig. 3 Transient absorption spectroscopy at 77 K. a, b** Transient absorption maps of BA$_2$FAPbI$_7$ following photoexcitation at 2.17 eV (**a**) and 2.43 eV (**b**) at 77 K. **c, d** Residual kinetics after subtracting a multi-exponent fit from the TA kinetics under pump energy of (**c**) 2.17 eV and (**d**) 2.43 eV, that were obtained by integrating around 2.15 eV.

2.43 eV, respectively. These maps are dominated by a positive $\Delta T/T$ bleach signal at 2.14 eV (red). When excess energy is provided by the excitation pulses (Fig. 3b), the positive $\Delta T/T$ peak is initially blue shifted and undergoes a rapid red shift over hot exciton cooling timescales of ~0.4 ps. Thus, we attribute it to hot-exciton cooling and the concomitant emission of phonons.

For both photoexcitation energies, we observe oscillations in the TA signal, with opposite phases on either side of the main photobleach peak (at 2.15 eV and 2.18 eV, see Supplementary Fig. 6). To extract the oscillatory component, we select the probe energy at which the oscillation amplitude is maximum and fit that trace using a combination of exponential and oscillatory components (see Supplementary Note 2 for details). The non-oscillatory components are subtracted from the raw data and the residual is shown in Fig. 3c, d for excitation energies of 2.17 and 2.43 eV, respectively. Similar oscillations were reported for a 2D perovskite variant in ref. [18] who attributed the effect to resonant impulsive stimulated Raman scattering (RISRS). We observe a cosine modulation of the TA signal which is consistent with both RISRS and displacive excitation of coherent phonons (DECP)[51,52]. DECP occurs when the equilibrium position of the ions experiences a sudden shift as they couple to the photoexcited charge distribution and gives rise to oscillations as the ions find their new minima[52]. Similar effects have been reported in the 3D perovskites[53]. This observation implies that photoexcitation drives crystalline distortion and indicates the presence of polaronic effects between the exciton and lattice. Interestingly, this implies that hot excitons can be generated by

photoexcitation of vibronic sub-levels without increasing **K**, consistent with our observations and discussion of the motional narrowing spin relaxation, which implied either that high **K** excitons do not exist for the linear relationship that is observed between scattering rate and spin lifetime, or that there is no net transport.

The oscillations occur with a wavenumber of 53 cm$^{-1}$ for both excitation energies (Supplementary Fig. 7), which is sufficiently low to fall within the bandwidth of the excitation pulses, and is similar in wavenumber to that reported for BA$_2$PbI$_4$[47]. As such, the differences in spin lifetime when BA$_2$FAPbI$_7$ is optically excited with and without excess energy cannot be explained purely from these impulsively generated cage distortions. Correspondingly, this leaves phonons emitted during hot-exciton cooling and the delocalisation of hot-exciton wave functions as likely drivers that change the dominant spin-relaxation mechanism, from motional narrowing to one which resembles that observed in 3D bulk perovskites. Notably, these hot-exciton properties can explain the differences in the spin-relaxation dynamics that we observe at room temperature between the two photoexcitation energies, as shown in Fig. 1d. Here, both datasets contain a fast, motional narrowing spin depolarisation of ~0.5 ps. However, the amplitude of this decay is reduced under 2.43 eV photoexcitation when more delocalised hot excitons are formed, which indicates a decrease in the number of states that experience motional narrowing spin depolarisation.

In order to confirm that optical injection of phonons is responsible for the formation of a polaronic state that undergoes a

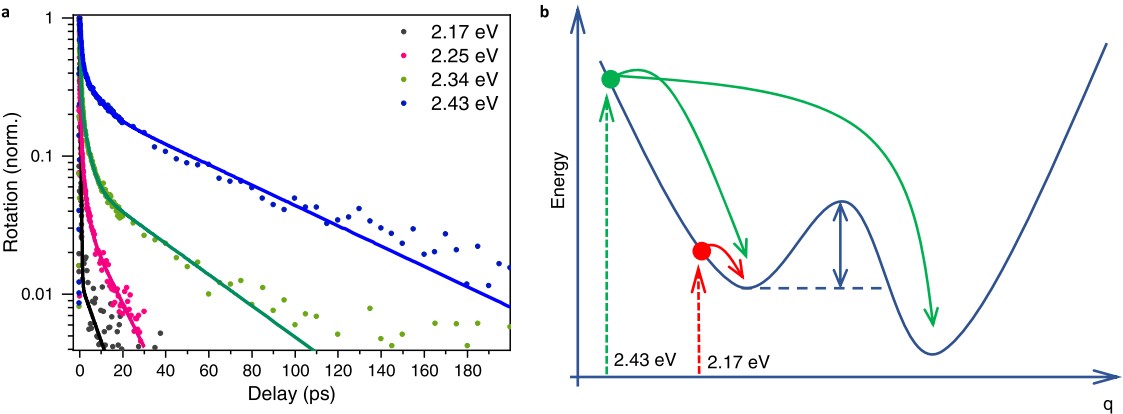

**Fig. 4 Faraday rotation with different photoexcitation energies at 77 K. a** Normalised Faraday rotation of $BA_2FAPbI_7$ at 77 K following photoexcitation at 2.17, 2.25, 2.34 and 2.43 eV. **b** Schematic showing the formation of a polaronic state following excitation at 2.43 eV, while photoexcitation at 2.17 eV preserves the exciton when no thermal energy is available.

different spin depolarisation mechanism, we perform FR at 77 K for different photoexcitation energies. Figure 4a shows that the spin lifetime $\tau_2$ increases as a function of the excess energy provided by the excitation photons. The contribution from the fast motional narrowing component, $\tau_1$ also decreases with excess energy. Together, these observations imply the formation of a new electronic state that requires additional energy to form, and which is dominated by a different spin-depolarisation mechanism. The formation of a polaronic state leads to a reduction in electron–hole wave function overlap, a process that relies on the availability of energy to overcome the Coulomb interaction.

We, therefore, propose that the formation of the polaronic state is a continuous that occurs across an activation barrier, a scheme of which is displayed in Fig. 4b. Photoexcitation at 2.17 eV fails to provide any excess energy to form polaronic states, and therefore only the spin-relaxation mechanism of the excitons is observed. By increasing the photoexcitation energy, an exploration of the excited state landscape is enabled, which slows down spin relaxation. The greater the excess energy, the greater the 'transfer rate' between exciton and polaronic states, and the greater the contribution from the polaronic population to the spin depolarisation time. In the limit of infinite excess energy, the observed spin depolarisation mechanism would become that of the polaronic states. As $\tau_2$ is our observable and as the starting population of excitons shows a 0.2 ps spin lifetime, we lack direct access to the transfer rate between excitons and polaronic states, since we always observe some spin relaxation while the excitons cross the activation barrier. This complicates the use of an Arrhenius plot to extract the activation barrier height, but we can extract an estimate of 18.4 meV, which is consistent with Fig. 2a (Supplementary Fig. 8). Incidentally, this discussion also explains the presence of the third component observed in Fig. 2d. Since spin relaxation is happening simultaneously along the formation of polaronic states, frustrated barrier crossings will lead to significant spin lifetime inhomogeneity with a transfer-rate-weighted-average lifetime between that of the excitons and that of the polaronic states.

## Discussion

Consequently, we attribute the new, slow spin depolarisation mechanism to the formation of polaronic states, for which the spatial separation between electron and hole wave functions is increased[40,54,55] and lattice symmetry modified. We propose that these changes will decrease the rate of spin precession that is expected within the MSS[40] or D'yakonov–Perel mechanisms, respectively, which in turn allows a new depolarisation mechanism

to dominate[50]. While the formation of polaronic states modifies many kinetics by also changing scattering rates, screening of the Frohlich interaction, and broadening the energy distribution, they do not impact the spin relaxation the way observed here (Supplementary Note 3). The requirement of excess excitation energy for the formation of polaronic states to occur can be attributed to the increased exciton delocalisation[22] and the corresponding reduction in electron–hole wave function overlap of hot states, which are more similar to the conformation of the polaron state. Specifically, the polaron configuration requires excess energy to overcome the Coulombic electron–hole attraction. Due to the strong dependence of the optical properties on small changes to the organic cations[29,31,48,56–58], our observations highlight the need for further studies of low-temperature spin depolarisation in the presence of strong exciton–phonon interactions within other layered perovskite variants.

Based on results from time-resolved Faraday rotation and transient absorption spectroscopy we have proposed that the formation of polaronic states radically modifies the dominant spin-relaxation mechanism in the $n = 2$ metal-halide perovskite $BA_2FAPbI_7$. We rationalise the change in dominant spin relaxation mechanism by a decrease in electron–hole wave function overlap[55] or change in lattice symmetry which act to reduce the rate of motional narrowing spin precession[40,54] that we observe for the exciton state. Consistent with other reports that contend exciton–phonon interactions[21], we observe that the optical properties of the exciton transition are strongly coupled to a $\approx 50\,cm^{-1}$ phonon mode[51], with distinct exciton–lattice coupling[9].

Our analysis of spin depolarisation as a function of temperature suggests that thermal energy is sufficient for the formation of polaronic states, which constitute a notable fraction of the photoexcited population at room temperature. Upon cooling to cryogenic temperatures, the proportion of thermally activated polaronic states quickly falls, while their spin lifetime increases. Therefore, by exciting far above the exciton resonance, low-temperature polaronic states with significantly longer spin depolarisation times can be generated. We have shown that the spin polarisation lifetime of these states decreases with increasing temperature, as is observed for free carriers in 3D perovskites. As this mechanism remains unclear, further low-temperature investigations should be performed to determine if there is a link to the rotation of the small intralayer organic cation. It has been shown that this cation gives rise to a dynamic disorder when the thermal rotation barrier is overcome[17] and that the rotation time of this cation is inversely dependent on temperature, being

very slow at low temperatures and ~3 ps at room temperature[59] reminiscent of the spin depolarisation reported here and observed in the 3D perovskites. Recent theoretical calculations[19] also suggest that the B-site cation plays a crucial role in polaron formation in 3D perovskites. These results demonstrate that the strong exciton–phonon coupling in 2D perovskites sets them apart as a unique material system which enables optical control over the dominant cryogenic spin-relaxation mechanism by manipulating the formation of polaronic states. Due to their inherent tunability, this material class provides a further means to control exciton and polaron spin transport and interaction mechanisms, which may hold promise for spintronic devices.

## Methods

**Sample preparation**. Polycrystalline films of 2D halide perovskites were fabricated following the general formula of $A_2B_{n-1}Pb_nI_{3n+1}$, where $A =$ butylammonium, $B =$ formamidinium, and $n = 2$. Lead iodide, butylammonium iodide, and formamidinium iodide are dissolved in dimethylformamide at 0.1 M concentration and ratio of 2:2:1. Glass coverslips were sonicated in first acetone, then isopropanol for 5 min, before 10 min of oxygen plasma etching. The precursor solution was then spin coated at 2000 r.p.m for 60 s, followed by annealing at 100 °C for 60 s inside the nitrogen filled glovebox. All chemicals were procured from Sigma-Aldrich.

**Faraday rotation measurements**. The transient Faraday rotation experiments were performed using a regenerative amplified Ti:Sapphire system operating at 2 kHz. The pump and probe beams were generated using homemade optical parametric amplifiers (OPA) both with 40 meV full-width-half-maximum bandwidth and 0.1 ps duration. A quarter waveplate immediately before the sample was used to circularly polarise the pump beam, while the probe is linearly polarised. The pump was chopped at 1 kHz, and the probe was detected using a pair of balanced photodiodes placed on a rotation mount after a Wollaston prism pair, such that by rotating the mount the linearly polarised probe can be perfectly split in half between the photodiodes at a negative pump-probe time delay. Finally, a lock-in amplifier was used to detect the difference between the signal at the photodiodes with the pump on and with the pump off, effectively measuring the rotation of the polarisation of the probe induced by the circularly polarised pump.

**Transient absorption measurements**. Transient absorption measurements are performed using a supercontinuum generated by tight focusing 800 nm pulses into a sapphire plate, and a narrowband pump from an OPA (tuned from 575 to 490 nm with 10 nm FWHM at 575 nm). The material is excited by the pump and the probe is passed through the excited region. The transmitted probe beam is dispersed in a monochromator and detected using a CCD array. The change in transmission, $\Delta T/T$, is calculated as a function of wavelength and pump-probe delay using $\Delta T/T = (T_{on} - T_{off})/T_{off}$, where $T_{off}$ and $T_{on}$ are the transmission signals of the probe when the pump has been blocked or unblocked by an optical chopper. The pump-probe delay is increased up to 1.5 ns using an automated delay stage.

## Data availability

The data that support the findings of this study are available on request from the corresponding authors (G.C. and F.D.). Source data are provided with this paper.

## Code availability

No special code was developed to fit the experimental data.

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

## Acknowledgements

S.A.B. acknowledges the support from the EPSRC Centre for Doctoral Training in Graphene Technology (EP/L016087/1). S.G. and G.C. acknowledge the Marie Curie actions (project H2020- MSCA-IF-2018-841356). T.W. and F.D. acknowledge funding from an EPSRC NI grant (EP/R044481/1). T.W. received funding from the European Union's Horizon 2020 research and innovation programme under Marie Skłodowska-Curie grant agreement no. 838772 (LADIE). F.D. acknowledges a Winton Advanced Research Fellowship and funding from the DFG Emmy Noether Program. T.W.J.G. acknowledges funding from the Schiff Scholarship and support from the EPSRC Cambridge NanoDTC, EP/L015978/1. R.S. acknowledge funding and support from the SUNRISE project (EP/P032591/1) and Newton International Fellowship from The Royal Society. G.C. acknowledges support from the European Union Horizon 2020 Programme under Grant Agreement 881603 Graphene Core 3.

## Author contributions

S.A.B., F.C., S.G., G.C. and F.D. planned the experiments that were carried out by S.A.B., F.C., S.G., T.N. and T.W.J.G. Samples prepared by T.N. and R.S. F.D., T.W. and G.C. supervised the project. S.A.B., F.C. and S.G. performed data analysis. S.A.B. and F.C. wrote the manuscript with feedback from all authors.

## Funding

## Competing interests

The authors declare no competing interests.
