## [Peer Review File · Nature Communications]

Optical control of exciton spin dynamics in layered metal halide perovskites via polaronic state formationREVIEWER COMMENTS

Reviewer #1 (Remarks to the Author):

The main finding of this work, namely that of a two orders of magnitude increase in the exciton spin lifetime at 77 K, when under photoexcitation with photon energy in excess of the exciton absorption peak, is very interesting for a variety of reasons, both fundamental and applied. I believe that the data presented in this work proves that this is indeed the case, although it also leaves questions unanswered (see list of questions below).

The part that I am not convinced by, is the effort to interpret the origin of this behavior. I recognize that this is primarily an experimental paper, and I appreciate the efforts to provide a theoretical scenario that is consistent with the findings, but I have trouble understanding what this is. Several relevant questions are listed below; I used the summary at the end to illustrate my lack of understanding of what is actually being proposed.

I believe that this part of the manuscript should be significantly improved, before it can be accepted for publication.

List of questions:

1. In the introduction (lines 46-50), the authors suggest the strong electron-phonon coupling + confinement in the 2D heterostructures result in the formation of exciton-polarons. This is consistent with many other works, cited there.

The conclusion (lines 180-181), however, attribute the newly observed slow dephasing rate to "formation of an exciton-polaron following hot exciton cooling".

Q: do the authors expect that exciton-polarons are generically present in these materials, or do they expect them to appear only under the special conditions used here (low-T, excitation high photon energies)? I am confused.

2. Lines 182-183 attribute this lower rate to a decreases of "the rate of spin precession within the MSS mechanism", which "allows the depolarisation mechanism of 3d bulk perovskites to dominate".

Why would a 3d mechanism (presumably for rather bare excitons) describe the behavior of 2D dressed excitons = excitons-polarons?

3. Line 183 speculates about "the requirement of excess excitation energy for polaron formation to occur". Surely, this could be verified experimentally by varying the pump energy with a smaller step, and showing more results in between the 2.17eV resonant energy, and the 2.43eV higher value studied here. (As an aside, how/why was 2.43eV energy chosen? We are not given any explanation).

If this scenario is correct, one would expect this increased lifetime to set in once the excess photon energy surpasses this polaron formation energy. Extracting such a polaron formation energy (if the data indeed corroborates this scenario) would be very useful for many purposes. More data to substantiate this would be very welcome.

4. Next line: "Specifically, the polaron configuration requires excess energy to overcome the Coulombic electron-hole attraction". To me, this suggests that the exciton-polaron is actually an unstable (Meta-stable?) higher-energy state, not one describing the exciton in these layers. This seems to contradict the statement in the introduction (question 1).

Perhaps part of the confusion comes from the fact that the authors do not explicitly define what they mean by exciton-polaron, maybe they attribute it a different meaning than I do (mine agrees with, eg, that discussed in ref. [18]).

More minor points:

1. why is amplitude of τ_2 disappearing below about 180K in Fig 2b, but not in Fig 2d? Is this τ_2 supposed to be the same in both figures? It has a fairly similar value but it's quite strongly T-dependent in (b) and rather constant in (d).

For the fits in (d), the authors have kept τ_1 like in (b). Why would that component stay unchanged but τ_2 would change between the two conditions? Is now τ_2 rather constant with T because its T-dependence is now taken over by τ_3 ? What is the power-law of the T-dependence of τ_2 in (b)? If it's similar to that of τ_3 in (d), this would suggest that perhaps these quantities have not been properly disentangled.

2. What is the phase transition that happens around 250K? It would be useful to say it, if it's known. Why is only τ_1 affected by it, but not τ_2 nor τ_3 ?

3. Is there a known phonon at/around 53 cm^{-1} energy? That would nicely compliment the data in Fig 3.

Minor issues:

Fig 3(b) should be labeled 2.43eV.

Reviewer #2 (Remarks to the Author):

The authors report a study of spin relaxation in the 2D perovskite (BA₂FAPbI₇). They report temperature-dependent measurements of Faraday rotation for two different pumping energies: (i) resonant with the exciton; and (ii) pumping 260 meV above the exciton. For resonant pumping, the trend vs temperature is consistent with motional narrowing, but for pumping configuration (ii), a slower spin relaxation process is observed with the opposite dependence on temperature. Since hot carrier cooling in pumping configuration (ii) would produce excess phonons in the sample (which they also see evidence for in supporting transient absorption studies), they conclude that phonons are impacting the mechanism of spin relaxation. This is an interesting result since a physical picture of dynamic disorder and electron-phonon coupling in the hybrid perovskites is just now starting to emerge. The authors postulate that the motional narrowing is due to electron-hole exchange that gets weaker once polarons are formed due to a larger electron-hole separation.

Several issues must be addressed before considering publication.

1. Polaron formation has many consequences that could affect the spin lifetime, such as a change in the scattering rate, exciton energy distribution, screening of the Frohlich interaction with other phonons, etc. The authors present an argument that their observations are due to an increase in the electron-hole separation within the dressed exciton that changes the strength of the exchange coupling, and that this change reduces the strength of the effective magnetic field within the MSS spin relaxation mechanism. This argument requires that the MSS mechanism dominates for undressed excitons, however the arguments for this dominance are somewhat weak, as discussed below. The other ways that polaron formation could change the exciton kinetics and the spin lifetime should also be included in the discussion. Is there another explanation involving polarons that is consistent with their findings?

2. The authors observe a spin lifetime that scales with the scattering rate. The authors argue that this is consistent with motional narrowing provided that the effective magnetic field involved is independent of temperature. The authors then suggest that "high-K excitons are unlikely to exist",

as justification for neglecting the temperature dependence of $\Omega(K)$, but a thermal population of excitons will possess a spread of K values even if the center of the distribution is at $K=0$ (i.e. there is no net transport). For instance, the dominance of DP motional narrowing in III-V semiconductors is due to the thermal spread of K and occurs in the absence of a net center-of-mass motion of electron-hole pairs. The discussion in this section is therefore problematic and should be modified accordingly.

3. The authors say that their experiments are carried out in the low exciton density regime, however only the pulse fluences are presented. A quantitative estimate of the exciton density in their experiments is needed.

4. The authors should discuss their findings in the context of the Bir-Aronov-Pikus mechanism, which is also due to electron-hole exchange.

A few more minor issues:

5. The reference to the spintronics literature is quite limited. For the benefit of the reader, I would suggest adding the following textbooks to the citation list: *Optical Orientation*, edited by F. Meir and B. P. Zakharchenya (Elsevier, Amsterdam, 1984) and *Semiconductor Spintronics and Quantum Computation*, edited by D. D. Awschalom, D. Loss, and N. Samarth (Springer, Berlin, 2002).

Reference to papers showing the utility of controlling the spin lifetime would also add useful context. For example, see *Phys. Rev. Lett.* 90, 146801 (2003), *Appl. Phys. Lett.* 83, 2937 (2003), and *Phys. Rev. Lett.* 91, 246601 (2003).

6. As the manuscript is currently structured, the primary conclusions are somewhat buried in the time-constant analysis. I found that the content of the section 4 within the Supplemental Material (with the title "Other") to be a clearer representation of the bigger picture implications of the work. I would recommend that this section be worked into the main document as part of the conclusion section.

Reviewer #3 (Remarks to the Author):

The authors study spin physics in layered $\text{Ba}_2\text{FAPbI}_7$ compound at low temperatures using a Faraday rotation and TA techniques. They find an increase of the spin-lifetime by two orders of magnitude at 77K with photon energy about that to exciton energy.

Here the authors studied spin relaxation when the photon energy is higher than the exciton energy and leads to the formation of exciton-polaron states.

When the photoexcite at a high photon energy the spin-relaxation dependence on temperature is different than when photoexcitation occurs resonant with the exciton energy.

These results are intriguing and I can support publication in Nature Communication.

The results are explained in terms of form exciton-polarons only when the excitation energy is in excess of the exciton energy.

Did the authors measure the effect at other excess energies? If the mechanism discussed in the text is correct then it would seem that one would see differences in the PL as a function of excess energy. Was that checked? Could the results be better explained by the formation of free-carriers

rather than exciton polarons? I suppose then the carriers would need to be localized within polaronic states prior to forming excitons.

REVIEWER COMMENTS

Reviewer #1 (Remarks to the Author):

The main finding of this work, namely that of a two orders of magnitude increase in the exciton spin lifetime at 77 K, when under photoexcitation with photon energy in excess of the exciton absorption peak, is very interesting for a variety of reasons, both fundamental and applied. I believe that the data presented in this work proves that this is indeed the case, although it also leaves questions unanswered (see list of questions below).

The part that I am not convinced by, is the effort to interpret the origin of this behavior. I recognize that this is primarily an experimental paper, and I appreciate the efforts to provide a theoretical scenario that is consistent with the findings, but I have trouble understanding what this is. Several relevant questions are listed below; I used the summary at the end to illustrate my lack of understanding of what is actually being proposed.

I believe that this part of the manuscript should be significantly improved, before it can be accepted for publication.

We thank the reviewer for the careful reading of our manuscript and the positive assessment of our core finding. We are also grateful to the reviewer for raising unclear points in the interpretation of the results. We have revised this part of the manuscript, following the guidance from the questions below.

List of questions:

1. In the introduction (lines 46-50), the authors suggest the strong electron-phonon coupling + confinement in the 2D heterostructures result in the formation of exciton-polarons. This is consistent with many other works, cited there.

The conclusion (lines 180-181), however, attribute the newly observed slow dephasing rate to "formation of an exciton-polaron following hot exciton cooling".

We thank the reviewer for this comment, which made us aware of issues with the wording used to describe the excited states in layered perovskites and has motivated us to clarify our language both in this response and in the manuscript.

In recent years several articles emphasized the strong exciton-phonon coupling in layered perovskites, introducing the nomenclature "exciton-polarons" and "dressed excitons" to refer to these excited states. In this nomenclature the word polaron emphasizes the role of lattice distortions in the excited state. The exciton-polaron is the strongly allowed (bright) state that is formed immediately following light absorption.

This nomenclature is in contrast to the word "polaron" as routinely used in the organic photovoltaic community, which refers to a charge transfer state that relies on distortions of the soft organic lattice to enable partial charge separation (see for instance [dx.doi.org/10.1021/cm4027144](https://doi.org/10.1021/cm4027144) | Chem. Mater. 2014, 26, 561–575). The latter use of the term thus refers to excited states that are not optically accessible from the ground state and can be formed following excitation of bright excitons or charge injection through an external electric current. There is however evidence for the presence of states that resemble

the latter “polarons” in perovskites, both from theory and experiment (some relevant references are listed below).

- Theory:
 - Adv. Energy Mater. 2020, 10, 1902748 DOI: 10.1002/aenm.201902748
 - PHYSICAL REVIEW LETTERS 127, 067401 (2021) doi: 10.1103/PhysRevLett.127.067401
- Experiment:
 - Adv. Mater. 2018, 30, 1707312 DOI: 10.1002/adma.201707312
 - PHYSICAL REVIEW LETTERS 122, 166601 (2019) DOI: 10.1103/PhysRevLett.122.166601
 - J. Chem. Phys. 152, 214705 (2020), DOI: 10.1063/5.0008608

We believe that the main difficulty of the original manuscript was that we had missed to make a clear distinction between the aforementioned states.

As the referee correctly points out, it is well-established that exciton-polarons are the states populated in layered perovskites following optical absorption. Instead, our data suggests that a polaronic state exists, but that its formation requires an activation energy. We explain this below and in the revised manuscript, whilst emphasizing the distinction between standard polaronic states and exciton-polarons. Therefore, this comment highlights that indeed there was lack of clarity in the language of the original manuscript.

The revised manuscript now cleans up the terminology and simply uses the terms exciton and polaronic state to distinguish our observations. This fully resolves this issue, and we thank the reviewer once more for helping us to identify this problem.

Q: do the authors expect that exciton-polarons are generically present in these materials, or do they expect them to appear only under the special conditions used here (low-T, excitation high photon energies)? I am confused.

Considering the reply to the comments above, we believe that exciton-polarons are the optically accessible states in these materials. To avoid confusion, we now use the term *exciton* to refer to the photoexcited bright state and *polaronic state* to refer to the new state that we find to form with excess energy and changes the spin depolarisation mechanism.

Our data strongly suggests that in these specific perovskites polaronic states, separated by an activation energy barrier, can also be formed after photoexcitation of excitons. The formation of any population of polarons thus depends on the availability of energy that is required to overcome this activation barrier. This energy may be provided either by the overall temperature or by the excess energy injected by the excitation photons.

The consequence of this new state is most obvious in the Faraday rotation data at 77K, where we observe a stark change in spin depolarisation kinetics. When exciting at 2.17 eV there is no thermal energy to form polaronic states, either from the bath or from the excess photon energy. Hence, the time-resolved Faraday rotation shows the spin-depolarisation time of the excitons at this temperature (~200 fs). However, when we photoexcite at 2.43 eV, carrier phonon scattering takes place on a timescale that is competitive with the spin-depolarisation of excitons. Consequently, a sizeable population of polaronic states is formed before spin depolarisation. The polaronic states that are formed before their exciton “parents” lose spin memory then show a significantly longer spin dephasing time

with *opposite temperature dependence* — a new spin relaxation mechanism now dominates. Since the spin lifetime observed at 77K following 2.43 eV excitation is much longer than that observed under any other condition, this result indicates the presence of a new, stable state.

Changes: to clarify our language, we made the following changes to the manuscript.

- “Within the layered metal halide perovskites, the term ‘exciton-polaron’ has been introduced to describe the coupling between excitons and lattice vibrations(Thouin, Srimath Kandada, *et al.*, 2019; Thouin, Valverde-Chávez, *et al.*, 2019; Duan *et al.*, 2020; Urban *et al.*, 2020), while further studies have indicated the formation of polaron states(Esmailpour *et al.*, 2020) which are not well described by the Fröhlich Hamiltonian(Ram, Kandada and Silva, 2020) and may require a significantly more complex theoretical treatment(Hong Sio *et al.*, 2019). For simplicity, in this paper we refer to the former simply as excitons and the later as polaronic states.”
- We have avoided our originally incorrect use of the term exciton-polaron to refer to the second state that forms when excess energy is provided. We now propose that this is simply a polaronic state. Our initial use of ‘exciton-polaron’ was intended to point out that we do not expect this to be a free charge coupled to the lattice, but rather a partially dissociated exciton stabilized by lattice distortions.
- To avoid confusion, we now use the term *exciton* to refer to the photoexcited bright state and *polaronic state* to refer to the new state that forms with excess energy and changes the spin depolarisation mechanism.

2. Lines 182-183 attribute this lower rate to a decreases of "the rate of spin precession within the MSS mechanism", which "allows the depolarisation mechanism of 3d bulk perovskites to dominate".

Why would a 3d mechanism (presumably for rather bare excitons) describe the behavior of 2D dressed excitons = excitons-polarons?

The spin relaxation mechanism for 3d perovskites is not yet fully understood, however the temperature dependence of the 3d mechanism obeys the same power law as that of the new state demonstrated here. Nevertheless, we agree with the referee’s primary point that we should not equate these mechanisms here.

While we have not been specific in the main text, we believe that there is a link between spin relaxation mechanisms that may originate from the small organic cation that is present in $n > 1$ 2d perovskites and 3d bulk perovskites. This is something that we are currently investigating. If correct, and this spin relaxation mechanism does exist, then it can be expected to manifest for all $n > 1$ 2d perovskites if the fast spin depolarization from motional narrowing is sufficiently slowed or eliminated.

In this revised manuscript, we propose an interpretation of why the spin relaxation from motional narrowing could be slowed enough for another mechanism to dominate: either as a consequence of reduced wavefunction overlap or change in lattice symmetry when a polaronic state forms. Thanks to the insights from the reviewers, we have clarified the state in question as polaronic, provided further discussion on the impact of polaron formation on spin relaxation, and moved the discussion from

section 4 of the SI into the main text to highlight a possible link between 3d and 2d spin relaxation mechanism.

Changes: to clarify the link between spin depolarization mechanisms in 2D and 3D perovskites, we added the following sentence:

- “Consequently, the dominant spin relaxation mechanism changes to one which resembles that of the 3d perovskites, and is yet to be determined” (Odenthal *et al.*, 2017).
- Taken from the SI and added to conclusion: “As this mechanism remains unclear, further low temperature investigations should be performed to determine if there is a link to the rotation of the small intra-layer organic cation. It has been shown that this cation gives rise to a dynamic disorder when the thermal rotation barrier is overcome (Sharma *et al.*, 2020) and that the rotation time of this cation is inversely dependent on temperature, being very long at low temperatures and ~ 3 ps at room temperature (Bakulin *et al.*, 2015)—reminiscent of the spin depolarisation reported here and observed in the 3d perovskites. Recent theoretical calculations (Duan *et al.*, 2020) also suggest that the B-site cation plays a crucial role in polaron formation in 3d perovskites.”

3. Line 183 speculates about "the requirement of excess excitation energy for polaron formation to occur". Surely, this could be verified experimentally by varying the pump energy with a smaller step, and showing more results in between the 2.17eV resonant energy, and the 2.43eV higher value studied here. (As an aside, how/why was 2.43eV energy chosen? We are not given any explanation).

If this scenario is correct, one would expect this increased lifetime to set in once the excess photon energy surpasses this polaron formation energy. Extracting such a polaron formation energy (if the data indeed corroborates this scenario) would be very useful for many purposes. More data to substantiate this would be very welcome.

We thank the reviewer for this suggestion. Indeed, supplementary Figure S6 of the original manuscript included an additional data point, but it was not sufficiently emphasized. Given that a similar point was also raised by reviewer 3, we opted to bring this data to the main text as a new Figure 4 (displayed here as Figure R1) alongside a scheme to explain how the polaron formation leads to the unusual spin lifetime dependence for these perovskites. Unfortunately, the bandwidth of our excitation pulse (40 meV) is so broad that the level of detail required to determine the onset energy is not possible to obtain via the method proposed by the referee.

Figure R1 | Faraday rotation with different photoexcitation energies at 77K. a Normalized Faraday rotation of $\text{BA}_2\text{FAPbI}_7$ at 77K following photoexcitation at 2.17, 2.25, 2.34 and 2.43 eV (dots) alongside multi-exponential fits (lines). **b** Scheme showing the formation of a polaronic state following excitation at 2.43 eV, while photoexcitation at 2.17 eV preserves the exciton when no thermal energy is available.

Unfortunately, the bandwidth of our excitation pulse (40 meV) is so broad that we cannot resolve the polaron formation energy via the method proposed by the referee. Instead, to probe the existence of an activation barrier, we must consider that polaron formation is a continuous process of electron and hole overcoming the Coulomb interaction and separating from one another in physical space and being stabilized by nuclear reorganization. Considering that our observable is spin lifetime and that the starting population of excitons shows a 0.2 ps spin lifetime, we lack direct access to the transfer rate between excitons and polaronic states, since we always observe some spin relaxation while the exciton crosses the activation barrier. This is aggravated by the fact that as charges are separating, the electron-hole overlap is gradually reduced, giving rise to some lifetime inhomogeneity that justifies the intermediate 3 ps component we observe. Nevertheless, we note that the polaron spin depolarisation time, τ_2 , scales with pump photon energy and, as such, best reflects the transfer rate between exciton and polaron states. We can therefore construct an Arrhenius plot, Figure R2, which shows the expected linear behaviour. The caveat is that the faster equilibration between excitons and polarons enabled by the extra phonons is observed through a slowing of the spin lifetime. Therefore, the resulting Arrhenius plot has a positive slope, but still reflects the crossing of the activation barrier.

Figure R2| Arrhenius plot of τ_2 as obtained in Figure 4 against the inverse excess energy provided by the excitation pulses. Standard analysis yields an activation barrier height of 18.4 meV, which may be skewed by the lifetime inhomogeneity linked to the simultaneous character of spin relaxation and polaron formation. We note that 18.4 meV corresponds to the thermal energy of 210K, the temperature range for which spin lifetime changes between mono and biexponential in Figure 2b. The unusual positive slope arises from the fact that our observable is the spin lifetime, which gets slower as the activation barrier crossing gets faster.

A naïve analysis of the Arrhenius plot in Figure R2 yields a barrier height of 18.4 meV, which is reasonable in comparison with reported phonon energies. Particularly, 18.4 meV correspond to the thermal energy available at 210K, which is the temperature range for which the Faraday rotation fits in figure 2b stop being monoexponential. However, since we cannot provide a rigorous argument regarding how precise this barrier height is, we prefer to leave this plot and the corresponding estimate in the supporting information.

As for the reason behind choosing 2.43 eV as the pump photon energy for the main experiments, it was just a convenient choice for our laser system that provided plenty of excess photon energy with respect to the bandgap.

Changes: We added Figure R1 to the main text as Figure 4 alongside the following paragraph:

“In order to confirm that optical injection of phonons is responsible for the formation of a polaronic state that undergoes a different spin depolarisation mechanism, we perform FR at 77K for different photoexcitation energies. Figure 4a shows that the spin lifetime τ_2 increases as a function of the excess energy provided by the excitation photons. The contribution from the fast motional narrowing component, τ_1 also decreases with excess energy. Together, these observations imply the formation of a new electronic state that requires additional energy to form, and which is dominated by a different spin depolarisation mechanism. The formation of a polaronic state leads to a reduction in electron hole wave function overlap, a process that relies on the availability of energy to overcome the Coulomb interaction.

We therefore propose that the formation of the polaronic state is a continuous that occurs across an activation barrier, a scheme of which is displayed in Figure 4b. Photoexcitation at 2.17 eV fails to provide any excess energy to form polaronic states, and therefore only the spin relaxation mechanism of the excitons is observed. By increasing the photoexcitation energy, an exploration of the excited state landscape is enabled, which slows down spin relaxation. The greater the excess energy, the greater the 'transfer rate' between exciton and polaronic states, and the greater the contribution from the polaronic population to the spin depolarisation time. In the limit of infinite excess energy, the observed spin depolarisation mechanism would become that of the polaron. As τ_2 is our observable and as the starting population of excitons shows a 0.2 ps spin lifetime, we lack direct access to the transfer rate between excitons and polaronic states, since we always observe some spin relaxation while the excitons cross the activation barrier. This complicates the use of an Arrhenius plot to extract the activation barrier height but we can extract an estimate of 18.4 meV, which is consistent with Figure 2a (Supplementary Figure 8). Incidentally, this discussion also explains the presence of the third component observed in Figure 2d. Since spin relaxation is happening simultaneously along the formation of polaronic states, frustrated barrier crossings will lead to significant spin lifetime inhomogeneity with a transfer-rate-weighted-average lifetime between that of the excitons and that of the polaronic states.

Further, we added Figure R2 to the supporting information as the new Supplementary Figure 8 with the additional discussion:

"As τ_2 is our observable and as the starting population of exciton-polarons shows a 0.2 ps spin lifetime, we lack direct access to the transfer rate between exciton-polaron and polaron, since we always observe some spin relaxation while the exciton-polaron crosses the activation barrier to form the polaron. This complicates the use of an Arrhenius plot to extract the activation barrier height. Nevertheless, we note that the spin depolarisation time τ_2 , scales with pump photon energy as it reflects the transfer rate (the ratio) between exciton and polaron states. We therefore construct an Arrhenius plot, Figure R2, which shows the expected linear behaviour with the caveat that an increase in transfer rate is observed through a *slowing* of the spin lifetime. Therefore, the resulting Arrhenius plot reflects the crossing of an activation barrier but has a very unusual positive slope.

Analysing the Arrhenius plot yields a barrier height of 18.4 meV, but this value will be skewed by the indirect nature of our observable. In any case, this value is consistent with the observation in Figure 2b that as the temperature is increased, the spin relaxation following 2.17 eV excitation stops being monoexponential at around 210K, which corresponds to a thermal energy of 18.4 meV."

4. Next line: "Specifically, the polaron configuration requires excess energy to overcome the Coulombic electron-hole attraction". To me, this suggests that the exciton-polaron is actually an unstable (Meta-stable?) higher-energy state, not one describing the exciton in these layers. This seems to contradict the

statement in the introduction (question 1).

Perhaps part of the confusion comes from the fact that the authors do not explicitly define what they mean by exciton-polaron, maybe they attribute it a different meaning than I do (mine agrees with, eg, that discussed in ref. [18]).

The referee is correct in that the confusion is due to the definition of the word “exciton-polaron”, and we hope that we have clarified it now. Since only the exciton-polaron state is accessible through optical excitation, immediately following high energy photoexcitation all excited states are exciton-polarons, which we are simply calling excitons. From there, a thermodynamic equilibration and its kinetics depend on the availability of phonons to form the polaronic state. If the excitation photons do not provide excess energy and neither does the bath at a given temperature, we will only observe spin relaxation from excitons. On the other hand, if excess energy is provided by the excitation photon, a corresponding equilibrium of excitons and polaronic states will form and the spin relaxation of both populations will be observed.

More minor points:

1. why is amplitude of tau_2 disappearing below about 180K in Fig 2b, but not in Fig 2d? Is this tau_2 supposed to be the same in both figures? It has a fairly similar value but it's quite strongly T-dependent in (b) and rather constant in (d).

For the fits in (d), the authors have kept tau_1 like in (b). Why would that component stay unchanged but tau_2 would change between the two conditions? Is now tau_2 rather constant with T because its T-dependence is now taken over by tau_3? What is the power-law of the T-dependence of tau_2 in (b)? If it's similar to that of tau_3 in (d), this would suggest that perhaps these quantities have not been properly disentangled.

We apologize for the inconsistency in our labelling. τ_2 in the original Fig. 2b is indeed τ_3 of Fig. 2d. What was labelled τ_2 in Fig. 2d is a new parameter and as such should have been called τ_3 – it was originally labelled τ_2 because it is the second longest component. We have corrected this confusing point so that τ_1 and τ_2 are the same in panels b and d.

In Figures 2b and 2d the transparency of each point is inversely proportional to the amplitude of each component, such that darker points correspond to large amplitudes. We note that for 2.17 eV excitation (Figure 2b) the polaronic state becomes less dominant at lower temperatures. This is because there is less thermal energy to overcome the activation barrier for polaron formation as discussed above. In Figure 2d we provide energy via the excitation photons, so we can observe how this parameter varies at low temperature, without its amplitude vanishing.

Changes: We swapped τ_2 and τ_3 across the main text and supplementary information such that τ_1 and τ_2 in Figure 2b and 2d correspond to the same physical quantities.

2. What is the phase transition that happens around 250K? It would be useful to say it, if it's known. Why is only tau_1 affected by it, but not tau_2 nor tau_3?

The thin film nature of our samples didn't allow for a detailed Rietveld assignment, so we cannot assign the different crystal phases at this point. Nonetheless, the presence of a phase transition is confirmed by XRD experiments (see Supplementary Figure 4). It could be the order-disorder phase transition that has been reported to occur at similar temperatures for BA_2PbI_4 , which could be indicative of polaron

formation and dynamic disorder due to the activation of the small organic cation, as discussed in the introduction of the main text.

τ_1 is the time constant connected to motional narrowing spin depolarisation. Importantly, we show from the PL linewidth that the scattering rate, Γ , changes in response to this phase transition and that the depolarisation time responds in unison, as expected by the formula below:

$$\tau_s \propto \frac{\Gamma}{\Omega^2(K)}$$

While τ_2 also depends on temperature, it is evident from Figure 2b of the main text that around 250K the value of τ_2 no longer changes significantly with changes in scattering rate (following $T^{-3/2}$). As such, changes in phonon scattering rate do not manifest as large changes in spin depolarisation time for this mechanism. Indeed, the data point at room temperature lies above our line of best fit, which would be in keeping with reduced phonon scattering, but we feel that this is within our error margins and cannot be concluded. Further, the temperature dependence of τ_2 is not consistent with any theory (it only follows the same power law as that observed in bulk samples which is yet to be explained), so its temperature dependence may not be coupled to changes in scattering rate. Indeed, the independence of τ_2 on the scattering rate could be consistent with our hypothesis that τ_2 is linked to the rotation of the FA, which has the same T-dependence (Bakulin *et al.*, 2015), for which further experiments are being performed.

τ_3 , on the other hand, seems to be related to spin relaxation that takes place as an exciton is attempting to cross the activation barrier to form polaronic states, so we do not expect it to be strongly affected by this phase transition.

Change to main text:

“Beyond 240K (grey regions of Figure 2b,d), the material undergoes a phase transition which may be linked to an order–disorder phase transition (Menahem *et al.*, 2021) as seen in BA_2PbI_4 (Supplementary Figure 4) that leads to a deviation from this power law.”

3. Is there a known phonon at/around 53 cm^{-1} energy? That would nicely compliment the data in Fig 3.

We thank the reviewer for pointing this out. Indeed, there was a recent report on a similar BAPbI_4 material, which shows pronounced Raman modes at 47 cm^{-1} . (Menahem *et al.*, ACS Nano, 2021, doi: 10.1021/acsnano.1c02022).

Change: We have added this reference now to the main text, along with the sentence:

“The oscillations occur with a wavenumber of 53 cm^{-1} for both excitation energies (Supplementary Figure 8), which is sufficiently low to fall within the bandwidth of both excitation pulses, and is similar in wavenumber to that reported for BA_2PbI_4 .”

Minor issues:

Fig 3(b) should be labeled 2.43eV.

Change: We thank the reviewer for the careful reading of the manuscript and have fixed this error.

Reviewer #2 (Remarks to the Author):

The authors report a study of spin relaxation in the 2D perovskite (BA₂FAPbI₇). They report temperature-dependent measurements of Faraday rotation for two different pumping energies: (i) resonant with the exciton; and (ii) pumping 260 meV above the exciton. For resonant pumping, the trend vs temperature is consistent with motional narrowing, but for pumping configuration (ii), a slower spin relaxation process is observed with the opposite dependence on temperature. Since hot carrier cooling in pumping configuration (ii) would produce excess phonons in the sample (which they also see evidence for in supporting transient absorption studies), they conclude that phonons are impacting the mechanism of spin relaxation. This is an interesting result since a physical picture of dynamic disorder and electron-phonon coupling in the hybrid perovskites is just now starting to emerge. The authors postulate that the motional narrowing is due to electron-hole exchange that gets weaker once polarons are formed due to a larger electron-hole separation.

We thank the reviewer for the careful reading of our manuscript and positive view on our main findings. We agree that a picture on the dynamic disorder and electron-phonon coupling in the hybrid perovskites is just now starting to emerge, and that their impact on exchange interactions was not yet considered. It appears to be highly relevant for electronic states due to the spin-orbit coupling in these materials.

Several issues must be addressed before considering publication.

1. Polaron formation has many consequences that could affect the spin lifetime, such as a change in the scattering rate, exciton energy distribution, screening of the Frohlich interaction with other phonons, etc. The authors present an argument that their observations are due to an increase in the electron-hole separation within the dressed exciton that changes the strength of the exchange coupling, and that this change reduces the strength of the effective magnetic field within the MSS spin relaxation mechanism. This argument requires that the MSS mechanism dominates for undressed excitons, however the arguments for this dominance are somewhat weak, as discussed below. The other ways that polaron formation could change the exciton kinetics and the spin lifetime should also be included in the discussion. Is there another explanation involving polarons that is consistent with their findings?

The reviewer correctly lines out our proposed picture for the observed changes in spin relaxation dynamics, which we had attributed to changes in the strength of exchange coupling following the formation of polaronic states. We have now added further discussion on other spin relaxation mechanisms in the manuscript and SI to clarify the picture for the reader. Importantly, we discuss the possible impact of the formation of polaronic states on scattering rates (see below) and we modify our discussion of the motional narrowing mechanism to account for both MSS or D'yakonov-Perel origin. Specifically, our original discussion for electron-hole overlap remains, as well as the potential loss of or reduction in lattice asymmetry that enables D'yakonov-Perel spin depolarisation.

A change in the exciton-phonon scattering rate would not change the mechanism in favour of the significantly slower spin depolarization at 77K. A decrease in scattering rate would increase the spin relaxation rate due to motional narrowing and lead to shorter lifetimes, contrary to our observation. In order to attribute the observed change in mechanism to scattering, the spin relaxation must be slowed down significantly, such that a new mechanism becomes the faster source of spin depolarisation.

However, our results in the main text (and those of Todd, Riley et al.) indicate that the spin lifetime at room temperature is well below 40ps. Therefore, while increasing the scattering rate can slow spin relaxation, it is not enough to extend lifetimes beyond 40ps, as is required for the new mechanism (that shows inverse temperature dependence) to dominate.

Screening of the Fröhlich interaction with other phonons would reduce phonon scattering. This would decrease the spin lifetime under the motional narrowing regime, while we see an increase after polaron formation. We therefore conclude that it is not applicable.

The energy distribution of polaronic states indeed tends to be broader than that of excitons at low temperatures since polaron formation does require interaction with phonons. However, following excitation at 2.43 eV the excitons themselves also interact with phonons and indeed we see that a fast component of 0.2 ps, similar to that observed in the 2.17 eV experiment, remains present. The intermediate component at ~3 ps indeed seems to be due to failed crossings of the activation barrier, which is an effect of the energy distribution. Nonetheless, it seems that the predominant mechanism is the formation of polaronic states and the change in the electron-hole overlap that this entails.

Changes: we added to the main text the following discussion:

“While the formation of polaronic states modifies many kinetics by also changing scattering rates, screening of the Frohlich interaction, and broadening the energy distribution, they do not impact the spin relaxation the way observed here (Supplementary Note 3).”

In Supplementary Note 3 we added the above discussion. In the discussion of D'yakonov-Perel spin depolarisation, we add the sentence:

“As \vec{B}_{eff} (and consequently $\Omega(\vec{K})$) depend on the potential gradient, $\vec{\nabla}V$, which originates from lattice asymmetry, changes in the lattice configuration can lead to changes in the rate of precession.”

2. The authors observe a spin lifetime that scales with the scattering rate. The authors argue that this is consistent with motional narrowing provided that the effective magnetic field involved is independent of temperature. The authors then suggest that “high-K excitons are unlikely to exist”, as justification for neglecting the temperature dependence of $\Omega(K)$, but a thermal population of excitons will possess a spread of K values even if the center of the distribution is at $K=0$ (i.e. there is no net transport). For instance, the dominance of DP motional narrowing in III-V semiconductors is due to the thermal spread of K and occurs in the absence of a net center-of-mass motion of electron-hole pairs. The discussion in this section is therefore problematic and should be modified accordingly.

We thank the reviewer for highlighting to us that the MSS spin relaxation mechanism depends on the average center of mass wavevector, $|\vec{K}|$, and that a large spread of K-values can exist alongside a small or zero value of $|\vec{K}|$.

Following the reviewer’s guidance, we have now rephrased the discussion in the main text to highlight that the average center of mass wavevector only scales with T in the case of large net transport, and (maintaining the original argument) that if it did, the spread of K-values has been argued to be small by others. For these reasons, we expect the precession term to be independent of T. We also deleted the

statement that high-K excitons cannot exist which was based on the discussion from another paper (Andrés Burgos-Caminalet al., 2020).

Changes: we changed the main text as follows:

- Within a motional narrowing regime, an increase in the temperature can lead to both a higher momentum scattering rate, Γ , and a faster spin precession, $\Omega(\mathbf{K})$, around an effective magnetic field if there is a non-zero average center of mass wavevector (Supplementary Note 1). The temperature dependence of these two contributions determines the temperature dependence of the total spin polarisation lifetime:
- “This result is consistent with previous studies which have shown cooling through a vibrational manifold of distinct exciton states (Straus *et al.*, 2016, 2020; Neutzner *et al.*, 2018; Ram, Kandada and Silva, 2020), and have suggested a narrow spread of K-values such that high-K excitons are unlikely to exist (Andrés Burgos-Caminal *et al.*, 2020). Interconversion within this manifold allows the average exciton energy to increase without a corresponding increase in exciton momentum. Alternatively, the temperature independence of $\Omega(\mathbf{K})$ can be explained by a zero average center of mass wavevector.”
- “Interestingly, this implies that hot excitons can be generated by photoexcitation of vibronic sub-levels without increasing \mathbf{K} , consistent with our observations and discussion of the motional narrowing spin relaxation, which implied that high-K excitons cannot exist for the linear relationship that is observed between scattering rate and spin lifetime.”
- “Interestingly, this implies that hot excitons can be generated by photoexcitation of vibronic sub-levels without increasing \mathbf{K} , consistent with our observations and discussion of the motional narrowing spin relaxation, which implied either that high-K excitons do not exist for the linear relationship that is observed between scattering rate and spin lifetime, or that there is no net transport.”

To the SI:

- “Elaborating on points from the previous section: energy follows a linear dependence on T (when averaging over the Fermi-Dirac distribution) so, when assuming a parabolic dispersion, $|K| \propto \sqrt{E}$. However, the precession term depends on the average center of mass wavevector, which can remain independent of, or weakly dependent on T. As such, in the case of temperature independent $|\overline{K}|$

$$\tau_S \propto \Gamma(T)$$

S7

and the spin relaxation time scales linearly with the momentum scattering rate.

3. The authors say that their experiments are carried out in the low exciton density regime, however only the pulse fluences are presented. A quantitative estimate of the exciton density in their experiments is needed.

We thank the reviewer for this suggestion and have provided the excitation densities now.

Changes: we added the relevant exciton densities for the low fluence Faraday rotation data in the introduction:

“Therefore, we perform all measurements at low fluences for which many body interactions do not play a significant role, corresponding to excitation densities of around 10^{16} cm^{-3} (see fluence dependent data reported in Supplementary Figure 2).”

We added the exciton density information to the caption of Figure 1:

“All measurements had an exciton density smaller than $5 \times 10^{16} \text{ cm}^{-3}$ (see Supplementary Figure S2).”

We also added the exciton densities corresponding to the largest and lowest fluences reported to the caption of Supplementary Figure S2:

“The exciton densities corresponding to the reported fluences are between 1.33×10^{16} and $1.55 \times 10^{19} \text{ cm}^{-3}$ in a; 0.29×10^{16} and $1.15 \times 10^{19} \text{ cm}^{-3}$ in b; 3×10^{16} and $0.79 \times 10^{18} \text{ cm}^{-3}$ in c; and 0.67×10^{16} and $1.2 \times 10^{18} \text{ cm}^{-3}$ in d.”

4. The authors should discuss their findings in the context of the Bir-Aronov-Pikus mechanism, which is also due to electron-hole exchange.

The Bir-Aronov-Pikus indeed is also due to the long-range electron-hole exchange interaction, however it is dominant in highly p-doped semiconductors below 10K and describes the case where a free electron is scattered by a hole via this long range exchange (see I. Zutic, J. Fabian, and S. Das Sarma, Rev. Mod. Phys., Vol. 76, No. 2, April 2004). This is the reason why we chose to discuss our results based on the MSS description, which is more promptly translatable to the undoped layered perovskite sample.

Changes: we have now added a discussion of the Bir-Aronov-Pikus process to the SI under spin relaxation mechanisms. The new text reads:

“As with the MSS mechanism, the BAP mechanism is due to the long-range exchange interaction between an electron and a hole. However, the BAP mechanism is dominant the case of doped samples, where excitonic states are unlikely to form due to the strong screening by the majority carrier population. Therefore, BAP describes the spin flip of a free electron upon scattering with a hole via the long-range exchange interaction. As BAFAPbI₃ is an undoped perovskite, this mechanism is not expected to play a significant role.”

A few more minor issues:

5. The reference to the spintronics literature is quite limited. For the benefit of the reader, I would suggest adding the following textbooks to the citation list: Optical Orientation, edited by F. Meir and B. P. Zakharchenya (Elsevier, Amsterdam, 1984) and Semiconductor Spintronics and Quantum Computation, edited by D. D. Awschalom, D. Loss, and N. Samarth (Springer, Berlin, 2002) .

Reference to papers showing the utility of controlling the spin lifetime would also add useful context.

For example, see Phys. Rev. Lett. 90, 146801 (2003), Appl. Phys. Lett. 83, 2937 (2003), and Phys. Rev. Lett. 91, 246601 (2003).

Changes: We thank the reviewer for these helpful suggestions and have added these references as numbers 1-5.

6. As the manuscript is currently structured, the primary conclusions are somewhat buried in the time-constant analysis. I found that the content of the section 4 within the Supplemental Material (with the title “Other”) to be a clearer representation of the bigger picture implications of the work. I would recommend that this section be worked into the main document as part of the conclusion section.

Changes: We thank the reviewer for the detailed reading of our manuscript and the constructive feedback for improvement. We have worked the Supplemental Material section into the Main manuscript conclusion section as suggested. The new conclusions read:

“Based on results from Using time resolved Faraday rotation and transient absorption spectroscopy we have proposed show that the formation of exciton-polaronic states radically modifies the dominant spin relaxation mechanism in the n=2 metal halide perovskite BA_2FAPb_3 . We rationalise the change in dominant spin relaxation mechanism by a decrease in electron-hole wave function overlap (Ambrosio *et al.*, 2018) or change in lattice symmetry which act to reduce the rate of exchange-driven motional narrowing spin precession (Maialle, De Andrada E Silva and Sham, 1993; He, Zhong and Scholes, 2010) that we observe for the exciton state. Consistent with other reports that contend exciton-polaron interactions (Thouin, Srimath Kandada, *et al.*, 2019), we observe that the optical properties of the exciton transition are strongly coupled to a $\approx 50\text{cm}^{-1}$ phonon mode⁵², with distinct exciton-lattice coupling⁹.

Our analysis of spin depolarisation as a function of temperature suggests that thermal energy is sufficient for the formation of polaronic states, which constitute a notable fraction of the photoexcited population at room temperature. Upon cooling to cryogenic temperatures, the proportion of thermally activated polaronic states quickly falls, while their spin lifetime increases. Therefore, by exciting far above the exciton resonance, low temperature polaronic states with significantly longer spin depolarisation times can be generated. We have shown that the spin polarisation lifetime of these states decreases with increasing temperature, as is observed for free carriers in 3D perovskites. As this mechanism remains unclear, further low temperature investigations should be performed to determine if there is a link to the rotation of the small intra-layer organic cation. It has been shown that this cation gives rise to a dynamic disorder when the thermal rotation barrier is overcome (Sharma *et al.*, 2020) and that the rotation time of this cation is inversely dependent on temperature, is very long at low temperatures and $\sim 3\text{ps}$ at room temperature (Bakulin *et al.*, 2015)—reminiscent of the spin depolarisation reported here and observed in the 3d perovskites. Recent theoretical calculations (Duan *et al.*, 2020) also suggest that the B-site cation plays a crucial role in polaron formation in 3d perovskites. These results demonstrate that the strong exciton-phonon coupling in 2d perovskites sets them apart as a unique material system which enables optical control over the dominant cryogenic spin relaxation mechanism by manipulating the formation of polaron states. Due to their inherent tunability, this material class provides a further means to control exciton-polaron spin transport and interaction mechanisms, which may hold promise for spintronic devices”.

Reviewer #3 (Remarks to the Author):

The authors study spin physics in layered BA₂FAPbI₇ compound at low temperatures using a Faraday rotation and TA techniques. They find an increase of the spin-lifetime by two orders of magnitude at 77K with photon energy about that to exciton energy.

Here the authors studied spin relaxation when the photon energy is higher than the exciton energy and leads to the formation of exciton-polaron states.

When the photoexcite at a high photon energy the spin-relaxation dependence on temperature is different than when photoexcitation occurs resonant with the exciton energy.

These results are intriguing and I can support publication in Nature Communication.

The results are explained in terms of form exciton-polarons only when the excitation energy is in excess of the exciton energy.

We thank the reviewer for their careful reading of our manuscript and the positive assessment of our results.

Did the authors measure the effect at other excess energies? If the mechanism discussed in the text is correct then it would seem that one would see differences in the PL as a function of excess energy. Was that checked? Could the results be better explained by the formation of free-carriers rather than exciton polarons? I suppose then the carriers would need to localized within polaronic states prior to forming excitons.

We thank the reviewer for this constructive suggestion. As replied to reviewer 1 above, the desired data was partially present in Supplementary Figure 6 of the original submission but presented with insufficient emphasis. This figure has now been complemented by a new excitation energy series which has been added to the main text as the new Figure 4, alongside a scheme to illustrate exactly what is meant by polaron formation (reported as Fig. R1 at page 4 above). An Arrhenius plot with the results was also added to the Supplementary Information as Supplementary Figure S8 (reported as Fig. R2 at page 5 above). We have preferred to perform pump photon energy dependent FR rather than PL experiments, for which analysis would be complicated by the question of the recombination mechanism and its impact on the observations.

Changes: see reply to reviewer 1, point 3.

REVIEWERS' COMMENTS

Reviewer #1 (Remarks to the Author):

I am satisfied with the extensive answers provided by the authors to all my questions, and the corresponding changes in the manuscript. I support the publication of this manuscript as is.

Reviewer #2 (Remarks to the Author):

I have reviewed the changes the authors have made to their manuscript in response to my comments as well as the comments of the other two reviewers.

The authors have responded satisfactorily to all my suggestions except point 2., which they seem to have misinterpreted. My point was that, in the motional narrowing mechanisms, the K inside $\Omega(K)$ is nonzero for a given exciton in the distribution even when the mean K value for the entire distribution is zero, as it always is unless the system is under bias. Any thermal distribution of excitons will contain a spread of values of K and thus we expect $\Omega(K)$ to be temperature dependent. Any argument relating to relaxation cannot eliminate this temperature dependence because there remains a spread of K values.

I don't feel strongly that this issue alone should prevent publication, as I am pleased with the other changes made to the revised manuscript in response to my comments.

I am a bit concerned that the edits made by the authors in response to some of the questions of reviewer 1 have made the manuscript somewhat less clear, although I will defer to that reviewer's opinion as to whether or not this is the case.

Reviewer #3 (Remarks to the Author):

The authors have addressed the concerns and issues that I brought up and I can recommend publication. While I still have many questions on the proposed mechanism and not entirely convinced I think the observation is quite interesting and should spur further investigations. Therefore, I can recommend publication of the revised manuscript.

REVIEWER COMMENTS

Reviewer #1 (Remarks to the Author):

I am satisfied with the extensive answers provided by the authors to all my questions, and the corresponding changes in the manuscript. I support the publication of this manuscript as is.

We thank the reviewer for the careful reading of our manuscript and for supporting its publication in Nature Communications.

Reviewer #2 (Remarks to the Author):

I have reviewed the changes the authors have made to their manuscript in response to my comments as well as the comments of the other two reviewers.

The authors have responded satisfactorily to all my suggestions except point 2., which they seem to have misinterpreted. My point was that, in the motional narrowing mechanisms, the K inside $\Omega(K)$ is nonzero for a given exciton in the distribution even when the mean K value for the entire distribution is zero, as it always is unless the system is under bias. Any thermal distribution of excitons will contain a spread of values of K and thus we expect $\Omega(K)$ to be temperature dependent. Any argument relating to relaxation cannot eliminate this temperature dependence because there remains a spread of K values.

I don't feel strongly that this issue alone should prevent publication, as I am pleased with the other changes made to the revised manuscript in response to my comments.

I am a bit concerned that the edits made by the authors in response to some of the questions of reviewer 1 have made the manuscript somewhat less clear, although I will defer to that reviewer's opinion as to whether or not this is the case.

We thank the reviewer for carefully reading our revised manuscript and for recommending its publication.

We agree with the reviewer that there will always be a distribution of K values, however we believe our analysis in Figure 2b shows that this is minor within the investigated temperature range.

Regarding the changes made in reply to the comments by reviewer 1, we feel that the current version is more accessible to a broader public while remaining conceptually equivalent to the previous.

Reviewer #3 (Remarks to the Author):

The authors have addressed the concerns and issues that I brought up and I can recommend publication. While I still have many questions on the proposed mechanism and not entirely convinced I think the observation is quite interesting and should spur further investigations. Therefore, I can recommend publication of the revised manuscript.

We thank the reviewer for their careful reading of our manuscript and for recommending its publication in Nature Communications. We agree with the reviewer that the physical mechanism underlying our observations is still not entirely understood and further studies will be required to completely describe this phenomenon.